# Gedit: Keyboard Gestures for Mobile Text Editing

Mingrui "Ray" Zhang*
University of Washington

Jacob O. Wobbrock†
University of Washington

## ABSTRACT

Text editing on mobile devices can be a tedious process. To perform various editing operations, a user must repeatedly move his or her fingers between the text input area and the keyboard, making multiple round trips and breaking the flow of typing. In this work, we present *Gedit*, a system of on-keyboard gestures for convenient mobile text editing. Our design includes a ring gesture and flicks for cursor control, bezel gestures for mode switching, and four gesture shortcuts for copy, paste, cut, and undo. Variations of our gestures exist for one and two hands. We conducted an experiment to compare *Gedit* with the *de facto* touch+widget based editing interactions. Our results showed that *Gedit*'s gestures were easy to learn, 24% and 17% faster than the *de facto* interactions for one- and two-handed use, respectively, and preferred by participants.

**Keywords**: Text entry; text editing; gestures; touch screen; ring gesture; smartphone; mobile devices.

**Index Terms**: Human-centered computing~Human computer interaction (HCI); Interaction techniques~Text input, gestural input.

## 1 INTRODUCTION

Text entry is a fundamental input task on almost all computers, including touch-based mobile devices like smartphones and tablets. However, while touch-based text *entry* has garnered much attention, touch-based text *editing* has garnered less. Text editing, the process of correcting text, moving and replacing the cursor, selecting character ranges, and performing operations like copy-and-paste, still largely borrows from desktop mouse interactions, leading to certain inefficient editing processes on touch-based mobile devices.

Modeless editing operations [18] such as copy, paste, and cut are often handled in a *touch+widget* [7] manner: to copy text, one must touch exactly on the text to be selected, long-press to trigger "selection mode," drag the selection endpoints to adjust the selection range, and then select "copy." However, the cursor is positioned using tap gestures, which are error prone because of the fat finger problem [20], especially when text characters are small [1]. Also, users must press long enough to exceed a time threshold to trigger selection mode, and later select "copy" in a popup menu to complete the operation. These extra steps significantly slow text editing on mobile touch screens. Moreover, if an editing operation must happen during the text entry process, one must lift one's finger from the keyboard area to directly interact with the text input area, introducing unnecessary round-trips [5,10,12] and breaking the flow of typing.

Previous work has demonstrated the feasibility of gesture shortcuts. Fuccella *et al*. [7] designed multiple gestures on the keyboard area for different editing operations. For example, one can perform a swipe gesture to move the cursor, or a "C" gesture to copy text. They further introduced a gesture to initiate editing mode to

---

*e-mail: mingrui@uw.edu
†e-mail: wobbrock@uw.edu

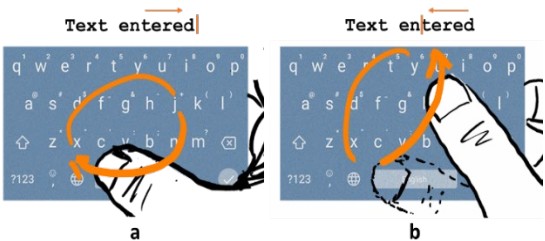

Figure 1. *Gedit*'s ring gesture for cursor control. (a) Draw clockwise to move the cursor continuously right, or (b) draw counterclockwise to move the cursor continuously left. Changing directions can be done in a single fluid gesture without lifting.

avoid conflict with gesture typing [8]. Building upon their work, we improve the cursor-moving and edit-initiating gestures and provide a gesture-only system, *Gedit*, for most text editing operations on mobile devices. For example, one of *Gedit*'s designs, the ring gesture, is shown in Figure 1.

Our work distinguishes itself from Fuccella *et al.* [7] and other prior work in four important respects: (1) instead of discrete cursor control (*e.g.*, one swipe gesture yields one cursor movement action), we provide a ring gesture for continuous, reversible cursor control. A significant advantage is that a user can move the cursor over a long range without clutching; (2) rather than using a single tap, we use bezel gestures [17] to enter editing mode, which we show is more distinguishable than a key-press [4]; (3) we added *undo* functionality to the gesture set, as *undo* is heavily used in text editing; and (4) we provide text editing gestures in both one- and two-handed modes, a significant design achievement given the constraints of one-handed use.

In our design of *Gedit*, we were careful to ensure that it remains compatible with gesture typing [13,22]. And because *Gedit* respects current interaction techniques, it is deployable on today's mobile systems without causing interference.

To evaluate *Gedit*, we conducted a text editing experiment. Our results show that compared to the *de facto touch+widget* method of text editing described above, *Gedit* was faster and preferred, especially for one-handed use.

## 2 GEDIT'S KEYBOARD EDITING GESTURES

*Gedit* contains ring and flick gestures for cursor control, bezel gestures for entering "editing mode," and letter-like gestures for copy, paste, cut, and undo. In editing mode, what was cursor movement becomes text selection. Editing gestures can be performed with one or two hands. We describe each of these interactions below.

### 2.1 Cursor Control

#### 2.1.1 Ring Gesture

The ring gesture is a rapid continuous reversible circular gesture [15,21] for moving the text cursor. The ring gesture offers three benefits over discrete cursor-control gestures: (1) users can continue drawing without resetting their finger position, avoiding clutching; (2) cursor speed can be adjusted as the gesture is being

performed; (3) users can change the cursor's movement direction without restarting the gesture.

The procedure for a user to perform the ring gesture is as follows: (1) The user begins to a draw a circular gesture. The drawing direction indicates the cursor movement direction: clockwise gestures move the cursor to the right; counterclockwise gestures move the cursor to the left (Figure 1a). (2) To change direction, the user reverses direction without lifting the finger (Figure 1b). The user can reverse repeatedly, enabling fine cursor control.

The text cursor moves as soon as the circle gesture is recognized. We used the `fitEllipse` function in the *OpenCV* library [3] to compute the ring center. We then use `fitEllipse` to update the ring center every 270° traversed by the gesture. This way, the center of the ring is dynamically updated during the drawing procedure, so the user can perform the gesture anywhere on the keyboard area, and the gesture can even "drift" while being drawn. The movement distance of the cursor is dictated by the angular distance traversed by the gesture. Through trial-and-error, we set the angular threshold of moving one character to 60°.

### 2.1.2 Flick Gesture

We implemented right and left flicks for word-level cursor movement, and up and down flicks for moving the cursor between lines of text, as shown in Table 1.

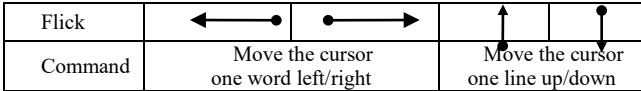

| Flick | ← • | • → | ↑ | • ↓ |
|-------|---------|---------|-----------|-----------|
| Command | Move the cursor one word left/right | | Move the cursor one line up/down | |

Table 1. Flick gestures for cursor control.

### 2.1.3 Gedit's Compatibility with Gesture Typing

Because *Gedit*'s text cursor-moving gestures require continuous contact with the keyboard, they could conceivably conflict with gesture typing systems like ShapeWriter [13,22]. If gesture typing is enabled, our solution is that only gestures starting on the spacebar key will be considered cursor-moving gestures. Flick gestures ending on the spacebar will also be considered as flick-down gestures. Given the large size of the spacebar, we found this solution to be both convenient and easy for users to perform.

## 2.2 Editing Mode

### 2.2.1 Entering Edit Mode via Bezel Gestures

To support other editing functions (*e.g.*, selection, copy, paste, cut, undo), we implemented a bezel mode-switch gesture for entering "editing mode" as shown in Figure 2. When the user swipes from the left edge of the keyboard, the keyboard becomes dimmed, indicating that editing mode has been entered. Then the user can perform different editing gestures with the other hand. To stop editing, the user simply lifts the finger that triggered editing mode. In editing mode, cursor movement gestures result in text selection.

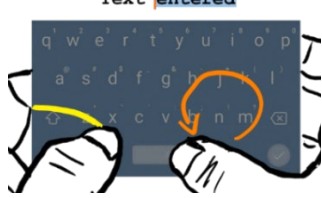

Figure 2. The left thumb swipes from the left edge to trigger editing mode, and then editing gestures can be performed by the right thumb. The ring gesture in editing mode performs text selection. To stop editing, the user simply lifts the left thumb.

---

### 2.2.2 Gedit's Gestures for Editing Commands

In editing mode, cursor control gestures will trigger character, word, or line selection. In addition, users can perform gestures for copy, paste, cut, and undo. The gestures are shown in Table 2.

| Gesture | C | V | ⋉ | Z |
|---------|------|-------|-----|------|
| Command | Copy | Paste | Cut | Undo |

Table 2. *Gedit* gestures for different editing commands.

We used the $P point-cloud stroke-gesture recognizer [19] to recognize the editing gestures. The $P recognizer is scale- and direction-invariant, which means that it works well regardless of the size, direction, and stroke sequence of a gesture. We designed the gestures according to their corresponding desktop shortcuts (Ctrl + C, V, X, or Z) for their mnemonic value, but any letter-like or symbolic equivalents could be defined. Because some gestures (such as "C") are similar to ring gestures, we included a 500 ms threshold to distinguish between the ring and other gestures in editing mode—if a gesture is performed quicker than this time threshold, it will be regarded as an edit gesture; otherwise, it will be regarded as the start of the ring gesture.

### 2.2.3 One-Handed Editing Gestures

*Gedit* also offers equivalent gestures for editing in one-handed mode. When holding and operating a phone in only one hand, using today's *de facto touch+widget* editing methods, the user has to uncomfortably stretch the thumb to reach the text input area. With *Gedit* for one-handed use, the editing mode-switch and editing commands are combined: the user first performs a bezel gesture from the right edge of the keyboard (for a right-handed user), and then fluidly *continues* to draw the editing gestures to make editing commands. Some examples of one-handed gestures are shown in Figure 3. To clarify, one-handed gestures can also be performed when holding with two-hands, and the user does not need to explicitly specify the mode between one- or two-handed modes.

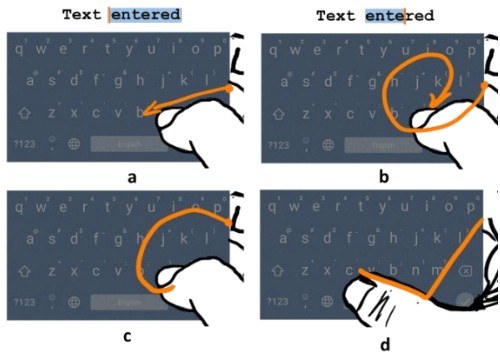

Figure 3. Some *Gedit* editing gestures in one-handed use. All gestures start from the right edge: (a) a flick left to select a word; (b) a clockwise ring gesture selects characters to the right of the text cursor; (c) the copy gesture "C"; and (d) the paste gesture "V".

## 3 GEDIT IMPLEMENTATION DETAILS

We implemented all *Gedit* gestures based on the Android Open Source Project (AOSP) keyboard [1] from Google. The keyboard provides a similar interface to the default Android keyboard with which many users are familiar. It also provides functions like statistical decoding for auto correction and word prediction. We sampled the gesture points at 50 Hz for our gesture recognition algorithms. We use the `vibrator` class of the Android system to

---

[1] https://android.googlesource.com/platform/packages/inputmethods/LatinIME/

implement haptic vibration feedback when the cursor moves. We also use a `toast` widget showing "Text copied!" to indicate the success of a copy gesture.

## 4 GEDIT EVALUATION

We conducted a text editing study to compare *Gedit* with the *de facto touch+widget* method. Specifically, we focused on evaluating efficiency and user preference.

We recruited 16 participants (aged 22 – 26, 6 male, 10 female) via email and word-of-mouth. All participants were right-handed and owned at least one touch screen device. Participants were compensated $15 USD for about an hour of their time. We made a text editor application for the experiment to log timestamps. A Google Pixel 2 XL smartphone was used for the study.

### 4.1 Study Design

To encourage participants to use *Gedit*'s gestures and editing features, we designed three sets of phrases for transcription. The phrases are shown in Table 3. As is evident, the phrases in each set contain a common string that is either rare (*e.g.*, "Tchaikovsky") or long (*e.g.*, "San Francisco"), which encouraged participants to use copy/cut and paste functions during typing.

| Experiment Phrases |
| --- |
| • Tchaikovsky was a Russian composer of the romantic period |
| • Tchaikovsky was educated for a career as a civil servant |
| • Tchaikovsky wrote many works that are popular |
| • With his music, Tchaikovsky was inspired to reach beyond Russia |
| • San Francisco is officially the city and county of San Francisco |
| • San Francisco is the traditional focal point of the San Francisco Bay Area |
| • San Francisco is known for its cool summers |
| • San Francisco is the highest rated American city on world livability rankings |
| • The polar bear is found in the Arctic Circle |
| • Methods of tracking polar bear populations are implemented |
| • The polar bear is an excellent swimmer |
| • Most polar bears are born on land |

Table 3. Phrases used in our transcription task. The phrases were selected from Wikipedia.

We conducted the study in both one- and two-handed modes. For each mode, the study was a within-subjects design, with a *Technique* factor having two levels: *Gedit* and *touch+widget*. The dependent variables analyzed in the study were completion time, editing time, and questionnaire scores. Editing time represented the time of performing editing operations—namely cursor moving, copy, paste, and cut—and was calculated from log files. For fairness between conditions, we did not include the undo function, because it was unavailable for *touch+widget*.

### 4.2 Procedure

Each participant had about 20 minutes to practice the gestures in both one- and two-handed modes. After practice, the experimenter presented the three sets of phrases, one set at a time, and told the participants to type them first using a desktop keyboard for familiarization, which reduced the effects of learning the phrases in the formal study. Then the participants started the formal transcription task using the smartphone. There were two blocks in the study: the first was to transcribe the phrases with two hands; the second was to transcribe the phrases with one hand. We counterbalanced the order of *Technique* in each block. Participants were then told to fill out a NASA TLX workload questionnaire [9,11] and to rate their preferences for each interaction in both one- and two-handed modes.

## 5 RESULTS

We gathered 192 sets of phrases for 768 phrases in total (2 handedness blocks × 2 techniques × 3 sets of phrases × 4 phrases per set × 16 people). We analyzed results for one- and two-handed conditions separately. We tested for order effects using linear mixed model analyses of variance [6,14], finding that condition order had no significant effect on any of our dependent variables, confirming the effectiveness of our counterbalancing.

Descriptive statistics for completion time and editing time are shown in Figure 4. For editing times, *Gedit* was 24% and 17% faster than the *touch+widget* technique for one- and two-handed use, respectively.

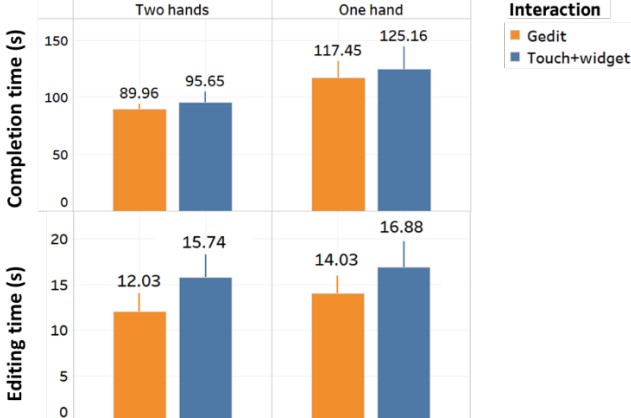

Figure 4. Completion time and editing time (lower is better) of *Gedit vs. touch+widget*. *Touch+widget* enables text editing by tapping on the keyboard and manually interacting with the text input area to position the cursor, make selections, etc. Error bars represent +1 standard deviation.

We log-transformed time values, as task-finishing times always follow the log-normal distributions [2] and performed paired-samples *t*-tests for both one- and two-handed conditions. For the one-handed condition, there was a significant effect of *Technique* on completion time ($t(15)=2.14$, $p<.05$), and a marginal effect on editing time ($t(15)=2.13$, $p=.050$). For the two-handed condition, there was a marginal effect of *Technique* on completion time ($t(15)=1.80$, $p=.092$), and a significant effect on editing time ($t(15)=3.97$, $p<.005$). This result indicated that *Gedit* gestures reduced editing and completion times compared to *touch+widget* editing.

The NASA Task Load Index (TLX) scores of two interactions are shown in Figure 5. We analyzed the average workload weighted score WWL (on a scale of ten) [9,16] of all six dimensions. Nonparametric Wilcoxon signed-rank tests revealed that when using one hand, editing with *Gedit* was perceived as significantly easier than editing with *touch+widget* ($Z=-2.36$, $p<.05$). However, when using two hands, there was no significant difference in perceived workload ($Z=0.13$, *n.s.*).

Preference ratings are also shown in Figure 5. The ratings ranged from 1 to 5, with 1 for the least preferred and 5 for the most preferred. Wilcoxon signed-rank tests showed that for one-handed use, the preference rating was significantly higher for *Gedit* than for *touch+widget* ($Z=3.42$, $p<.001$). But for two-handed use, there were no significant differences in the preference ratings of both methods ($Z=1.44$, *n.s.*).

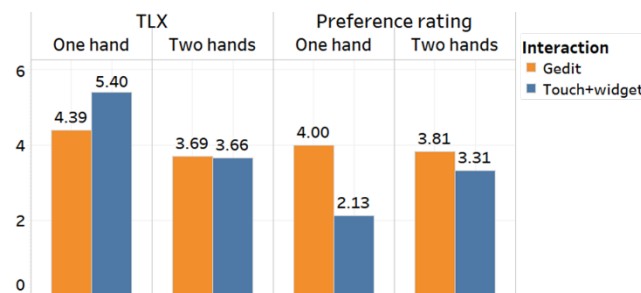

Figure 5. Mean NASA TLX workload ratings and preference ratings (1-5). (Lower TLX ratings and higher preference ratings are better.)

## 6 DISCUSSION

Our goal was to evaluate *Gedit* on its editing efficiency and users' subjective preferences. The results showed that our gesture interactions sped up the text editing process compared to the *de facto* editing approach of tapping keys and tapping the text input area to position the cursor (*touch+widget*). Participants especially appreciated the capability that *Gedit*'s gestures offered for one-handed use.

Participants generally enjoyed the *Gedit* gestures. The major reasons were having a feeling of precise control, convenience, and speed. Many participants also mentioned that the editing gestures such as copy and paste were faster than pointing and holding, and also less tedious to perform.

Participants had split preferences on the one- *vs.* two-handed versions of *Gedit*. Four participants preferred two-handed *Gedit* because "*it is more intuitive just like the shortcuts*" (P13), and "*the gestures in double-hand mode are easier to perform*" (P14). Four participants preferred one-handed *Gedit* because "*it is faster, as I don't need to enter the editing mode with another finger*" (P7). As for different gestures, we noticed that participants usually used the ring gesture to fix typos, while using the flick gestures to select whole words.

## 7 CONCLUSION

In this paper, we presented *Gedit*, a system of on-keyboard gestures for text editing: ring and flick gestures for cursor control and text selection, bezel gestures for mode switching, and letter-like gestures for editing commands. The gestures can be performed in both one- and two-handed modes. Through our formal study, we demonstrated that *Gedit* sped up the editing process and reduced text entry time, was perceived to require less workload, and was preferred to the *de facto* method of tapping keys and tapping text input areas. We also provided solutions to make our gesture interactions compatible with existing gesture typing techniques. Because *Gedit* is simple and easy to implement, it is easy to deploy on today's smartphones and tablets. It can co-exist with current methods, and can act as a complementary tool for those who desire a more efficient mobile text editing experience.

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
