# OpenReview forum: "Gedit: Keyboard Gestures for Mobile Text Editing"
_graphicsinterface.org/Graphics_Interface/2020/Conference — GI 2020_

### Official Review · AnonReviewer2 · 2020-01-06
**Interesting design, some issues with the study**

**Confidence:** 4
**Rating:** 6

**Review:**

The paper presents a new set of touch gestures to perform seamless transition between text entry and text editing in mobile devices. The authors expose their design rationales and the corresponding technique, then report a controlled experiment in which their technique was tested against a baseline in one- and two-handed text input tasks. Participants were overall faster with the candidate technique, which was preferred by participants in the one-handed condition.

The technique is well motivated and reasonably well described. I appreciate the addition of Undo to mobile text editing, and the frequent update of the ellipse parameters for the wheel gesture. The 500-ms delay is however kind of a bummer, but if the participants did not complain, then why not.
I am overall positive about this submission, but there are elements that would need to be clarified before I fully commit to it.

----------

# Study

p. 3: "For fairness between conditions, we did not include the undo function, because it was unavailable for touch+widget."
>> I actually have a problem with this. "Undo" with Gedit could in some case replace basic commands like backspace, which would then be counted for T+W but not for Gedit. Results without Undo could be reported to illustrate a point, but only in addition to the overall results (with Undo).

Why wasn't "number of hands" counterbalanced? This experiment protocol systematically "pre-trains" for one-hand in each condition. In particular, the results presented in the paragraph before [Conclusion] could be due entirely to order effects.
Plus, in the absence of counterbalancing, "number of hands" cannot be tested for order effect, and therefore the sentence "condition order had no significant effect on any of our dependent variables" (p. 4) is incorrect.

Why not compare the candidate technique to Fuccella et al.'s [7, 8]? This paper claims to build upon their work, but right now we don't know if the candidate technique actually does improve upon it. At the end of this paper, an interaction designer wouldn't know if they should pick [8] or this technique for their new system.


# Technique

How does one perform right-, up-, and down-swipes in the one-handed mode? Sounds like rather drastic angles are needed, e.g. left from the edge then backwards for a right swipe, that might not be correctly classified by an out-of-the-box recognizer. These specific gestures could use some more discussion and illustration for the one-handed case.
The inverted 'V' (right to left) also seems counter-intuitive, but for the user.


# Related work

This paper is missing some relevant references, notably regarding gesture augmentation of virtual keyboards ([A] and the work that followed), and circular gestures to avoid clutching [B, C].

[A] Jessalyn Alvina, Joseph Malloch, Wendy Mackay. Expressive Keyboards: Enriching Gesture-Typing on Mobile Devices. In UIST 2016.
[B] Grham Smith, m. c. schraefel, and Patrick Baudisch. Curve dial: eyes-free parameter entry for GUIs. In CHI ’05 EA.
[C] Sylvain Malacria, Eric Lecolinet, and Yves Guiard. Clutch-free panning and integrated pan-zoom control on touch-sensitive surfaces: the cyclostar approach. In CHI ’10.


# Clarity

p. 1: "cursor movement action" is unclear at this stage.

p. 2: "(4) we provide text editing gestures in both one- and two-handed modes, a significant design achievement given the constraints of one-handed use."
>> At this stage we don't know what the gestures are, and have little reason to believe that they should be different with one or two hands; the "significant design achievement" is not yet assessable (and perhaps a bit overclaimed?).

p. 3: "We conducted the study in both one- and two-handed modes. For each mode, the study was a within-subjects design, with a Technique factor having two levels"
>> This is a strange formulation, as it seems to imply that "modes" is not a within-subject variable (the text later confirms that it was).

Fig. 4: I know SD is symmetric, but the error bars should also be readable within the bar chart.
Fig. 5 should also display error bars.

Fig. 4: Why explain Touch+widget again in the caption?

The notion of "marginal effect" seems used rather generously (p>0.09), the paper should explain what made (and did not make) the cut.
Also, result sections should also report effect sizes, e.g. as differences of means.

---

### Official Review · AnonReviewer3 · 2020-01-07
**Neat solution to a frustrating problem.**

**Confidence:** 5
**Rating:** 9

**Review:**

This paper reports the design and evaluation of the Gedit interaction techniques. Gedit is a gesture based interaction on keyboard for cursor placement, text selection and text edition (copy, cut, paste, undo). Gedit can be used in a one or two handed manner. It was compared to the standard interaction technique used in Android Phones, and showed better performances and preferences in the one-handed condition compared to the standard interaction technique, while the two-handed condition had similar results but importantly not worst.

I would like to start my review by commending the authors for their work.
This paper is easy to read, flows nicely, is precise and concise.
The contribution is a neat technique that works well, is easy to implement in existing OSs. It tackles a concrete problem and provide a simple yet elegant and efficient solution.
The length is appropriate for the contribution.

I usually am lengthy in my reviews but after reading the paper, I only have two suggestions, which are no major concerns:
 - adding an explicit sentence in the "Editing Mode" section stating that once the editing mode has been entered every cursor movement results in a text selection operation (as pressing shift on a computer keyboard would do).
 - adding a short sentence explaining why authors "log-transformed time value" avoiding referencing back to [2].

I would therefore argue for accepting this paper. The topic is relevant and this work is well executed.

---

### Official Review · AnonReviewer1 · 2020-01-09
**Neat little system that seems to work well with both WGKs and Tap typing**

**Confidence:** 4
**Rating:** 8

**Review:**

While it is absolutely true that there has been a lot of work in gestures to control carat position on soft keyboards (Fuccella et al. have a great review of past techniques for discrete carat movement) and Fuccella et al.'s work includes a similar interaction for edit commands (cut/copy/paste). To me, this is a really nice piece of follow-on work that avoids the need to clutch multiple times to perform gestures, that integrates well with a WGK (unlike Fuccella where, afaik, assumes tap-based typing), and has a nice use of bezel to access command mode.

So, at a high level, this represents a  well-engineered enhancement to past work in this area, and I recommend its acceptance. Keeping with the positive, the prose is great, with no typos I noticed and no grammar issues either.

While the interaction technique is, in my opinion, neat and well-engineered, I have two concerns with the paper.

One concern with the paper itself has to do with the evaluation. I assume that the edit gestures could co-exist with traditional touch-screen carat placement? I assume that participants could skip copy-paste and just keep on typing? I would have loved to see how frequently participants used the edit gestures, what an optimal use of edit gestures (for select/copy/paste) would have been, and how close participants came to this optimal. To be clear, I don't consider this important from the perspective of user preference, but it would let me know what percentage of participants used these edit gestures and how frequently. Unless I missed it, it may be that edit time was relatively rare? Are there any participants that just transcribed? Was there something in the participant brief that pointed to copy-paste editing to ensure participants used this option, or was it self-directed?

A second (larger) concern with the paper came to light for me when I read the phrase, directly under Table 3, and then I went back to look at the one hand and two hand modes. Essentially, I am a bit confused by what one-handed and two-handed modes are. Unless I missed something, I am unclear whether these are explicit modes, or do they co-exist? Clarifying: my definition of mode follows Raskin's definition (Humane Interface): moded systems provide a different interpretation of an identical action given the state of the system. From this perspective, I understand what edit mode is, but I don't exactly understand whether one-hand vs two-hand is a separate mode or is it just a separate input constraint? I think it is the former (i.e. in one-handed mode, edit mode becomes an explicit mode, whereas in two-handed mode, edit mode is a quasi-mode, again as per Raskin, meaning that a bezel swipe gesture in one-hand mode behaves differently than in two-hand mode, but I could be wrong). I would really wish for some clarification on this.  This is a larger concern for me because, if you need to explicitly switch between one and two hand style keyboard input, it means that you have this moded keyboard. I'm actually not even sure it is needed; might it be possible to enter the mode if the user users a bezel swipe from the bottom of the keyboard (the most comfortable for one-hand bezel swipe) and a quasi-mode otherwise?

In summary, because of the novelty of the technique, I think this paper is a clear accept for Graphics Interface. If it were ever considered below the bar for a conference, I might suggest:
- That the authors clarify the one-hand vs two-hand modes (this should be done anyway) so that the design has good clarity. I admit that it is possible that I missed something in the paper about whether handed use was an explicit mode, but I don't think so. I have read and reread the section on design, and I don't think I missed anything. If there were a rebuttal period, this is something I would look for in rebuttal.
- That the authors consider a slighly different experiment where the phrases are transcribed with errors and the participants need to reposition the cursor and perform explicit editing operations to increase the frequency of the use of the gesture-based commands. It may be the case that some additional detail regarding experimental procedure would alleviate these concerns for me, simplifying the acceptance of the study data. Again, additional details are something I would request in a rebuttal to enhance the paper.

Smaller details:
- The log-transformed time was for the obvious reason of highly skewed data, I assume? Perhaps articulate those. Also, how skewed was the time, and was this a result of inaccurate typing or of more or less use of edits?

---

### Meta-Review · Area_Chair1 · 2020-01-14

**Recommendation:** Accept
**Confidence:** 5

**Metareview:**

All reviewers are in agreement that this paper represents an innovative approach to cursor placement and editing during text entry on soft keyboards.

While R3 has few concerns -- mainly of a minor typographical nature -- other reviewers note some potential problems with the experimental validation, including (R2):
- Lack of undo
- Counterbalancing
- Baseline (Fuccella's might have been better)
My own read of the paper leads me to believe that baseline could be addressed by a small expansion of the discussion of the shortcomings of Fuccella's work in regards to modern soft keyboards and WGKs, that the edit to correct for overstatement is a minor revision, and that the lack of undo, while a fair comment, does not rise to the level of invalidating the potential contribution of this work.

R2's concerns about technique clarification, and R1's concerns regarding moding, can primarily be addressed via an editing pass. I encourage the authors to revise their manuscript as needed to address these reviewer concerns.

---

### Decision · Program_Chairs · 2020-01-12

Accept